# HMVLM: Human Motion-Vision-Lanuage Model via MoE LoRA

**Lei Hu**[1,2*]          **Yongjing Ye**[1*]          **Shihong Xia**[1,2†]

[1]Institute of Computing Technology, Chinese Academy of Sciences
[2]University of Chinese Academy of Sciences
`{hulei19z, yeyongjing, xsh}@ict.ac.cn`

## Abstract

The expansion of instruction-tuning data has enabled foundation language models to exhibit improved instruction adherence and superior performance across diverse downstream tasks. Semantically-rich 3D human motion is being progressively integrated with these foundation models to enhance multimodal understanding and cross-modal generation capabilities. However, the modality gap between human motion and text raises unresolved concerns about catastrophic forgetting during this integration. In addition, developing autoregressive-compatible pose representations that preserve generalizability across heterogeneous downstream tasks remains a critical technical barrier. To address these issues, we propose the Human Motion-Vision-Language Model (HMVLM), a unified framework based on the Mixture of Expert Low-Rank Adaption(MoE LoRA) strategy. The framework leverages the gating network to dynamically allocate LoRA expert weights based on the input prompt, enabling synchronized fine-tuning of multiple tasks. To mitigate catastrophic forgetting during instruction-tuning, we introduce a novel *zero expert* that preserves the pre-trained parameters for general linguistic tasks. For pose representation, we implement body-part-specific tokenization by partitioning the human body into different joint groups, enhancing the spatial resolution of the representation. Experiments show that our method effectively alleviates knowledge forgetting during instruction-tuning and achieves remarkable performance across diverse human motion downstream tasks.

## 1 Introduction

With the capability of encoding semantic information and emotional expression, 3D human motion plays a critical role in virtual reality, embodied intelligence, computer graphics and visions. Recent advances in foundation language models [1, 61, 58, 23] have facilitated multimodal integration. This has stimulated researchers' interest in embedding 3D human motion into these models to address diverse motion-centric tasks, including text-to-motion synthesis[32, 76, 67], motion video understanding[5] and pose estimation [16]. Notably, M3GPT [44] develops a unified vocabulary that integrates text, motion, and music modalities, supporting both text-driven and music-driven motion generation applications.

Although prior work has made progress in motion-centric multimodal modeling, two key issues remain underexplored. First, the effect of incorporating human motion modalities on the foundation

---

*These authors contributed equally to this work.

†Shihong Xia is the corresponding author.

39th Conference on Neural Information Processing Systems (NeurIPS 2025).

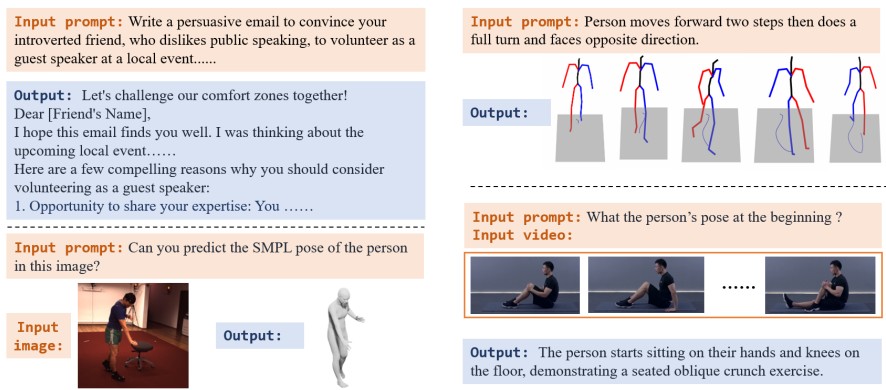

Figure 1: HMVLM preserves the original knowledge and dialogue capabilities of the foundation model while supporting a wide range of human-centric downstream tasks.

model's world knowledge is unclear. Dou et al. [12] observed that supervised fine-tuning improves model's instruction-following capabilities with expanding training data, yet simultaneously induces parameter deviation from pre-trained weights, progressively eroding pre-existing knowledge. Although approaches such as temporal continual learning [57] can mitigate catastrophic forgetting in unimodal motion tasks, the substantial modality gap between human motion and text necessitates a deeper examination of their compatibility within the foundation model. Otherwise, catastrophic forgetting may reduce the model to a task-specific generative system with limited dialogue abilities.

Second, how to formulate discrete motion representation compatible with autoregressive architectures in foundation models remains an open research question. Prior methods typically apply temporal convolution to extract motion features along the temporal axis and utilize VQ-VAE architecture to obtain discrete tokens. However, this tokenization paradigm overlooks the spatial information of the pose, limiting the expressiveness of single-frame representations in tasks like pose estimation. Therefore, developing spatially-aware and semantically-grounded tokenization methods for both motion sequences and static poses thus becomes imperative.

To address the first challenge, we observe that supervised instruction-tuning tends to overly focus on new tokens (e.g., motion-related tokens), causing the model to gradually forget its original world knowledge. Therefore, we introduce the Mixture of Expert LoRA framework (MoE LoRA) for multimodal fine-tuning. This framework aims to build a robust Human Motion-Vision-Language Model (HMVLM) for diverse human-centric downstream tasks (as shown in Fig. 1). The gating network dynamically routes task instructions to multiple LoRA expert pairs (LoRA_A/LoRA_B), enabling task-specific adaptation. To avoid knowledge forgetting, we further propose a non-trainable *zero expert* with zero-initialized parameters. We encourage the gating network to select *zero expert* for motion-unrelated tasks, thus preserving the pretrained weights of the foundation model and preventing catastrophic forgetting.

To solve the second issue, we segment the human body into distinct body parts and employ spatial transformers to encode each of them separately. This part-wise encoding, inspired by patch-based tokenization in image processing [11], enhances the resolution of motion or pose tokens while maintaining computational efficiency.

Experimental results show that the proposed HMVLM, built on the MoE LoRA framework, significantly reduces the model's forgetting rate while achieving strong performance in text-to-motion generation, monocular pose estimation, and motion video understanding. The main contributions of this work are:

1. We propose HMVLM, a unified framework that simultaneously supports multiple motion-relevant tasks, including text-to-motion generation, pose estimation, and motion video understanding.

2. We introduce the MoE LoRA architecture for multimodal and multitask fine-tuning of HMVLM, incorporating the novel concept of a *zero expert* to mitigate catastrophic forgetting and preserve foundational knowledge.

3. We design body-part-based tokenizers for pose and motion, improving representation granularity and boosting downstream task performance.

## 2 Related Work

### 2.1 Human Motion Modeling.

Deep learning methods have been extensively employed in conditional human motion generation and modeling tasks. These include deterministic motion prediction from historical states [17, 80, 18], motion completion [25, 54, 48], and motion control [26, 72, 55, 56, 69]. Additionally, deep learning approaches are widely utilized in human pose estimation tasks from RGB images or videos [3, 41, 37, 63, 39, 19, 13, 34]. With growing demands for diverse 3D human motions, probabilistic generation methods have emerged as a prominent research direction[42, 51, 66, 24, 27]. Moreover, leveraging large-scale and uniformly formatted datasets to pre-train a prior model [51, 18] has been shown to be an effective approach for handling multiple motion-related tasks. Among them, probabilistic text-to-motion (T2M), which involves learning cross-modal mappings between textual descriptions and 3D motions, is most related to our work. Early T2M methods focused on aligning modalities by creating a shared representation space for textual and motion features [47, 59]. MotionCLIP [59] embeds motion features into the CLIP latent space using rendered motion images. These methods faced limited motion diversity due to small datasets. Significant advancements followed the release of large-scale datasets such as KIT Motion-Language [49] and HumanML3D [21]. Recent methods including MDM [60], MotionDiffuse [74], ReMoDiffuse [75], and MLD [6] adopted diffusion models for text-guided motion generation, operating in original or compressed motion spaces. Another group of T2M methods employs VQ-VAE to embed motion into discrete latent embeddings, subsequently generating motion sequences autoregressively through Transformer-based architectures. Representative models include TM2T [22], T2M-GPT [73], AttT2M [79], and MoMask [20]. In addition, several works extend T2M approaches with motion editing capabilities, enabling the generation of motions that satisfy both textual descriptions and user-defined geometric constraints [35, 52, 8, 2]. While these methods have achieved impressive results in specific tasks, they primarily focus on human motion modeling or cross-modal learning rather than building robust multimodal frameworks capable of supporting multiple downstream tasks.

### 2.2 Foundation Models and Multi-modal.

Recent advances in foundation language models like ChatGPT [1], BERT [10], Llama [61], Gemma [58], and DeepSeek [23] have shown strong performance in language tasks, with excellent understanding, text generation, and adaptability. These advancements have laid a solid foundation for multimodal research, where instruction tuning has become a central focus, giving rise to frameworks such as vision-language models [43, 64, 46, 40], audio-vision-language models [71, 68], etc.

In the context of human motion, Jiang et al. [32] introduced MotionGPT, which treats 3D human motion as a "foreign language" and constructs a unified vocabulary through motion tokenization to support tasks such as text-to-motion and motion prediction. Zhang et al. [76] adopted Llama-2 as the base model and applied LoRA-based fine-tuning without modifying the word embeddings and prediction head. MotionChain [33] and MotionAgent [67] further enhance motion generation and understanding via multi-round instructions and GPT4-based coordination, respectively. MotionGPT-2 [65] extends MotionGPT by incorporating hand motion to enable whole-body motion generation. Most recently, M3GPT [44] integrates text, motion, and music modalities into a unified framework, supporting diverse cross-modal generation tasks. Although these studies have made progress in integrating human motion into foundational language models, their effects on the models' pre-trained world knowledge remain unexplored.

### 2.3 MoE LoRA Fine-tuning.

Mixture of Experts (MoE) [30] follows a divide-and-conquer strategy by routing inputs to specialized experts, and has been applied across various domains [14, 53, 72, 42, 80]. In foundation models, architectures like Switch Transformers [15] and DeepSeekMoE [9] leverage sparse routing to expand model capacity without increasing inference cost. Recent work [70, 38, 45, 12] integrates MoE with LoRA, showing it can match full fine-tuning performance while enhancing generalization [12].

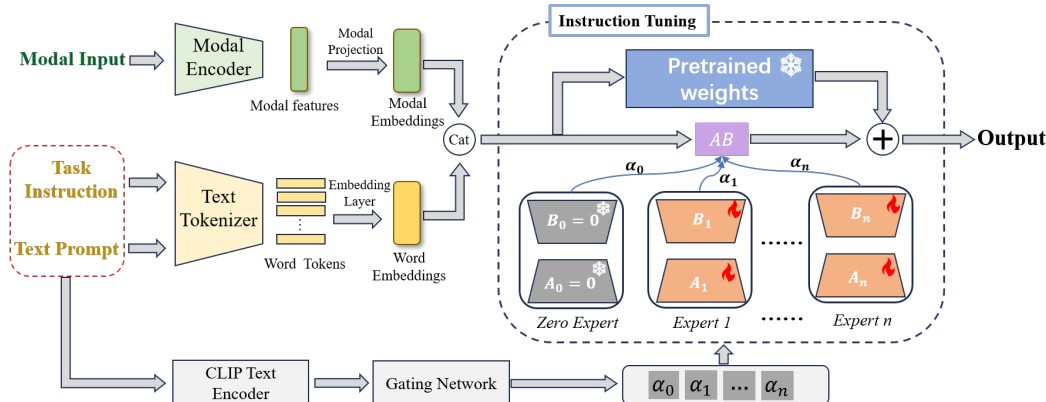

Figure 2: Method overview: task instructions and input prompt are processed by a gating network to produce a mixture weights. Modality-specific inputs are aligned with word embedding via projection layers, and the final outputs are generated through the pre-trained model and the weighted combination of LoRA experts.

Buehler et al. [4] further apply MoE LoRA to bioinformatical tasks such as materials analysis and protein design. Building on these insights, we apply MoE LoRA to HMVLM fine-tuning. Unlike prior approaches that place experts in Transformer feed-forward layers, we introduce multiple LoRA matrix pairs and employ a gating network to route instructions, enabling the model to preserve base model knowledge while adapting to diverse motion-related tasks.

## 3 Method

The overall framework of the proposed Human Motion-Vision-Language Model, based on MoE LoRA, is illustrated in Figure 2. The task instructions and text prompts are encoded using the CLIP text encoder and passed to the gating network $\omega$, which subsequently produces a mixture of expert weights $\boldsymbol{\alpha} = [\alpha_0, \alpha_1, ..., \alpha_n]$. Simultaneously, modality-specific inputs (e.g., image, video, or motion sequences) are projected into the foundation model's embedding space. The modal embeddings are then combined with the word embeddings and fed into the foundation model. Guided by the semantics of the task instructions and prompts, the LoRA experts are dynamically combined according to the computed weights $\boldsymbol{\alpha}$, enabling task-specific modulation.

### 3.1 LoRA Mixture

We modulate the foundation model's pretrained weights $W$ using multiple LoRA experts:

$$W' = W + \sum_{i=0}^{n} \alpha_i A_i B_i \tag{1}$$

Here, $n$ represents the total number of experts, while $A_i$ and $B_i$ are the corresponding LoRA matrices. We introduce a special *zero expert*, with non-trainable matrices $A_0$ and $B_0$ which are initialized to zero. When the gating network assigns a high weight to $\alpha_0$(i.e., approaching 1), the *zero expert* helps preserve the pre-trained parameters $W$, thereby mitigating catastrophic forgetting. Beyond knowledge preservation, the *zero expert* serves as a shared, general-purpose expert across tasks. Its weight $\alpha_0$ indicates the task's reliance on the foundation model's knowledge, enabling dynamic knowledge fusion and enhancing the synergy between the model and downstream tasks. This design thus provides flexibility and robustness in multimodal, multitask learning.

### 3.2 Multimodal Instruction Format

Given a pre-trained foundation language model $f_\phi(\cdot)$, where $\phi$ denotes the pre-trained parameters, the objective of this work is to construct a HMVLM $f_\psi(\cdot)$ leveraging MoE LoRA and instruction

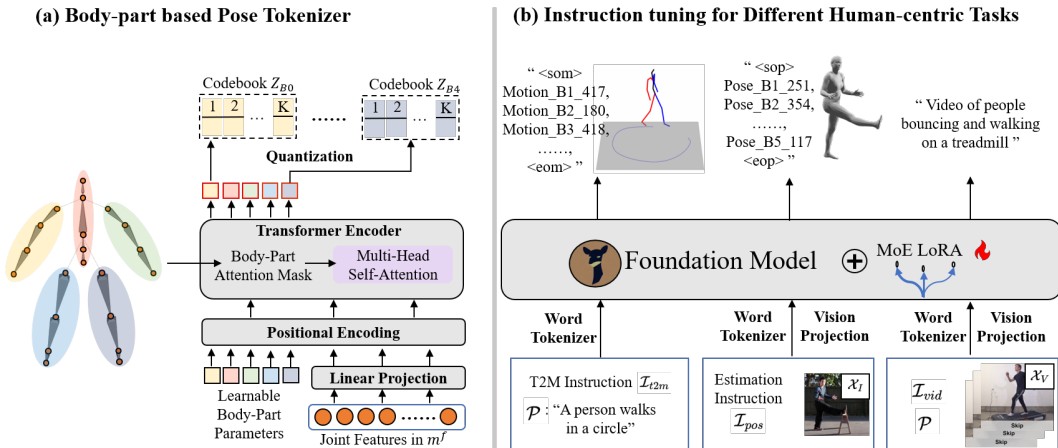

Figure 3: (a) Pose/motion tokenization scheme, we introduce learnable body-part parameters into the Transformer to facilitate feature pooling and quantization; (b) instruction tuning for diverse human-centric tasks. The discrete tokens are added to the foundation model's vocabulary, and then instruction tuning guides the model in generating task-related tokens.

tuning. The resulting model $f_\psi$ should not only retain the foundation model's original capabilities and knowledge but also effectively adapt to a diverse set of downstream tasks related to human motion. The general formulation of instruction tuning is as follows:

$$\mathcal{R} = f_\psi(\mathcal{I}, \mathcal{P}, \mathcal{X}) \tag{2}$$

where $\mathcal{I}$ denotes the task instruction, $\mathcal{P}$ represents the input prompt, $\mathcal{X}$ is an optional modality-specific input and $\mathcal{R}$ represents the model responses.

**Text-to-motion generation**: For this task, the model input-output formulation is $\mathcal{R} = f_\psi(\mathcal{I}_{t2m}, \mathcal{P})$, where $\mathcal{I}_{t2m}$ specifies a task-related instruction(e.g., "an AI assistant generates a motion sequence based on user description") and $\mathcal{P}$ contains a concrete user prompt (e.g., "a person walks clockwise in a circle."). The response $\mathcal{R}$ will then be encouraged to contain motion-specific tokens, which can be translated into 3D motion sequences by the motion decoder.

**Pose estimation**: For this task, the formulation is $\mathcal{R} = f_\psi(\mathcal{I}_{pos}, \mathcal{X}_I)$, where $\mathcal{X}_I$ is an image input and $\mathcal{I}_{pos}$ provides the task instructions for pose estimation. The output $\mathcal{R}$ contains pose-relevant tokens, which are then used by the pose decoder to infer the human pose from the input image.

**Motion video understanding**: For this task, the formulation is $\mathcal{R} = f_\psi(\mathcal{I}_{vid}, \mathcal{P}, \mathcal{X}_V)$, where $\mathcal{I}_{vid}$ is the instruction specific to human motion video understanding. $\mathcal{P}$ represents the input prompt and $\mathcal{X}_V$ is the video input.

### 3.3 Multimodal Instruction Tuning

To support pose estimation and T2M tasks, we will pre-train a pose and motion tokenizer (detailed in Sec. 3.4) to discretize the encoding of poses and motions (See Fig. 3 (b)), obtaining the pose vocabulary $V_m$ and the motion vocabulary $V_M$. These vocabularies are merged with the original text vocabulary $V_T$ to form an extended vocabulary $V = [V_T, V_M, V_m]$, while preserving the original text token order.

As illustrated in Fig. 3 (b), MoE LoRA enables joint fine-tuning across multiple human-centric tasks. Given instruction data of the form $(\mathcal{I}, \mathcal{P}, \mathcal{X}, \mathcal{R}_{gt})$, all tasks share the objective of next-token prediction:

$$\mathcal{L}_{fm} = -\mathbb{E}_{R_{gt}^t \in V}[\log p(\mathcal{R}_{gt}^t | \mathcal{I}, \mathcal{P}, \mathcal{X}, \mathcal{R}_{gt}^{<t})] \tag{3}$$

where $\mathcal{R}_{gt}^t$ denotes the ground-truth token at position $t$ in the response sequence and $\mathcal{X}$ is an optional modality-specific input, as described in Sec. 3.2. For image input $\mathcal{X}_I$, we use the pre-trained CLIP ViT-L/14 [50] with the Llava projection layer [43] to align visual features with the model's embedding space. For video input $\mathcal{X}_v$, 8 frames are uniformly sampled and processed similarly.

To preserve the foundation model's world knowledge in motion-unrelated tasks, we supervise the gating network $\omega$ using user prompts from conversation datasets. Specifically, the instruction $\mathcal{I}$ and user prompt $\mathcal{P}$ are first encoded by the CLIP text encoder, and then input into the gating network $\omega$ to obtain the expert weights $\boldsymbol{\alpha} = [\alpha_0, \alpha_1, ..., \alpha_n]$. To encourage selection of the *zero expert* ($\alpha_0$) in motion-unrelated tasks, we design the loss:

$$\mathcal{L}_{gat} = -\mathbb{E}[\eta * \log p_w(\alpha_0 | \mathcal{I}, \mathcal{P})] \tag{4}$$

Here, $\eta$ is an indicator function such that $\eta = 1$ if the input $(\mathcal{I}, \mathcal{P})$ is unrelated to human motion, and $\eta = 0$ otherwise. This loss helps retain the foundation model's linguistic capabilities and avoid catastrophic forgetting. For human motion-relevant tasks, the gating network dynamically combines experts to enhance the performance of downstream tasks. The final instruction tuning loss is given by $\mathcal{L}_{total} = \mathcal{L}_{fm} + \mathcal{L}_{gat}$.

### 3.4 Pose and Motion Tokenizer

To obtain the vocabularies $V_m$ and $V_M$, and integrate the pose and motion modalities into the foundation language model, prior works [73, 79, 32, 76] typically use VQ-VAE to discretize motion sequences. Specifically, given a motion sequence $M^{1:F} = [m^1, m^2, ..., m^F] \in \mathbb{R}^{F \times D}$, a motion tokenizer $\mathcal{E}$ applies 1D convolutions along the temporal axis to produce latent features $\hat{z}^{1:(F/l)} = \mathcal{E}(M^{1:F})$ ($l$ denotes the temporal compression ratio), followed by quantization:

$$z_i = \mathcal{Q}(\hat{z}_i) := \arg\min_{z_k \in Z} \|\hat{z}_i - z_k\|^2 \tag{5}$$

Here, $Z = \{z_i\}_{i=1}^{K} \subset \mathbb{R}^S$ is the learned codebook containing $K$ discrete latent vectors, each of dimension $S$. The full tokenization process is thus expressed as $z^{1:(F/l)} = \mathcal{Q}(\mathcal{E}(M^{1:F}))$. However, this classical tokenization focuses solely on temporal encoding by combining discrete codes across time, which limits its capacity to capture spatial granularity. In tasks like pose estimation, where only single-frame input is involved (i.e. $F = 1$), the accuracy of discrete encoding relies entirely on the fixed codebook. As a result, its ability to represent pose variations is severely constrained by the codebook size $K$, leading to coarse-grained representations.

To address this limitation, we draw inspiration from patch-based image encoding for spatial modeling [11]. We exploit the body's natural decomposition into limb-based parts. Part-aware modeling has proven effective in motion retargeting [28], motion style transfer [31], and text-to-motion [79], yet most prior work does not compute discrete codes independently for each part.

In this work, we adopt the spatial Transformer architecture from [28] and build body part-based pose and motion tokenizers, with the architecture illustrated in Fig. 3 (a). For a given pose $m^f$, we process each joint feature using a linear projection and go through the positional encoding with the learnable body-part parameters. During the self-attention computation, an attention mask matrix is constructed based on the correspondence between joints and body parts, ensuring that the body part parameters are only associated with joints within that part. Finally, we keep only the outputs corresponding to the body part parameters for pooling the pose embeddings. The spatial modeling process can be formulated as:

$$[\hat{z}_{B1}^f, \hat{z}_{B2}^f, ..., \hat{z}_{BN}^f] = \mathcal{E}_s(m^f) \tag{6}$$

where $N$ denotes the number of body parts. For single-frame pose input, a separate codebook is constructed for each body part, and the embeddings are quantized independently as $z_{Bn}^f = \mathcal{Q}_n(\hat{z}_{Bn}^f)$. For motion sequences, the spatial embeddings from $\mathcal{E}_s$ are further compressed along the temporal axis using a temporal convolution module $\mathcal{E}_t$, yielding:

$$(\hat{z}_{Bn}^{'1}, \hat{z}_{Bn}^{'2}, ..., \hat{z}_{Bn}^{'F/l}) = \mathcal{E}_t(\hat{z}_{Bn}^1, \hat{z}_{Bn}^2, ..., \hat{z}_{Bn}^F) \tag{7}$$

The tokenization processes for pose and motion are expressed as $\mathcal{Q}_{1:N}(\mathcal{E}_s(m^f))$ and $\mathcal{Q}_{1:N}(\mathcal{E}_t(\mathcal{E}_s(M^{1:F})))$, respectively. Following tokenization, separate decoders $\mathcal{D}_m$ and $\mathcal{D}_M$ are employed to reconstruct the original input as detokenizers. We adopt the training strategy of T2M-GPT [73], with the loss function $\mathcal{L}_M = \mathcal{L}_{rec} + \mathcal{L}_{emb} + \lambda_{com}\mathcal{L}_{com}$. Specifically, $\mathcal{L}_{rec}$ is the reconstruction loss and $\mathcal{L}_{emb}$ is used to update the codebooks, while $\mathcal{L}_{com}$ encourages the body part embeddings to remain close to their assigned codebook vectors.

# 4    Experiments

**Implementation Details.** We use Vicuna-7b-v1.5 [7] as the foundation language model with five LoRA experts (including a *zero expert*), each of rank 8. LoRA adapters are applied to all linear modules, and the gating network is implemented as a two-layer MLP with a hidden dimension of 512. It takes the 512-dimensional text features output by the CLIP model and predicts the weights for five experts. For detailed implementation, please refer to the Appendix.

**Datasets.** We train the gating network with the LMSYS-Chat-1M dataset [77], using 80% of the data for training. For the text-to-motion task, we use HumanML3D [21] and KIT-ML [49] datasets. Notably, the motion tokenizer is trained on the same training splits of HumanML3D and KIT-ML for consistency. For pose estimation and pose tokenizer training, we use the Human3.6M [29] and 3DPW [62] datasets. The MoVid dataset [5] is used for instruction tuning in motion video understanding.

## 4.1    Evaluation on the knowledge preservation

We assess how human motion modalities affect the model's knowledge retention. Specifically, we measure model forgetting by comparing text comprehension performance before and after text-to-motion instruction tuning, using the MT-Bench [78]. MT-Bench evaluates 80 questions across eight topics, each with two-turn dialogues. GPT-4 serves as the judge, scoring responses from 1 (completely incorrect) to 10 (fully correct). We use this to measure performance changes after T2M fine-tuning.

Table 1: Evaluation on dialogue abilities of foundation models before and after text-to-motion tuning

| Methods | FM | Write | Role | Extract | Reason | Math | Code | Stem | Code | Avg |
|---|---|---|---|---|---|---|---|---|---|---|
| MotionGPT [65] | Llama2 | 2.70 | 3.63 | 2.25 | 4.00 | 1.50 | 1.15 | 3.50 | 3.10 | 2.73 |
| | Tuned | 1.85 | 2.75 | 1.55 | 2.50 | 1.45 | 1.20 | 3.00 | 2.58 | 2.11 |
| MotionAgent [67] | Gemma2 | 8.83 | 8.65 | 7.90 | 7.25 | 5.80 | 5.30 | 8.93 | 9.70 | 7.79 |
| | Tuned | 1.00 | 1.00 | 1.00 | 1.00 | 1.00 | 1.00 | 1.00 | 1.00 | 1.00 |
| Ours | Gemma2 | 8.83 | 8.65 | 7.90 | 7.25 | 5.80 | 5.30 | 8.93 | 9.70 | **7.79** |
| | Tuned | 8.45 | 8.55 | 7.85 | 6.75 | 5.75 | 5.20 | 8.20 | 9.45 | **7.53** |
| Ours | Vicuna | 7.43 | 7.52 | 5.21 | 4.90 | 3.69 | 2.68 | 6.98 | 9.0 | **5.90** |
| | Tuned | 7.75 | 6.20 | 5.80 | 4.50 | 3.00 | 2.45 | 6.45 | 8.15 | **5.54** |
| Ours w/o $\mathcal{L}_{gat}$ | Vicuna | 7.43 | 7.52 | 5.21 | 4.90 | 3.69 | 2.68 | 6.98 | 9.0 | 5.90 |
| | Tuned | 1.00 | 1.00 | 1.00 | 1.00 | 1.00 | 1.00 | 1.00 | 1.00 | 1.00 |

We conduct a quantitative comparison between our method and two representative baselines, MotionGPT [32] and MotionAgent [67], with results summarized in Table 1. Since these methods are built on different foundation models, absolute MT-Bench scores are not directly comparable. Therefore, we primarily focus on the relative degradation in dialogue performance of each foundation model after instruction-tuning for the text-to-motion task.

As shown, MotionGPT, which applies LoRA solely to the *Query* and *Value* matrices without introducing new motion tokens, preserves part of the original linguistic capability but still exhibits a noticeable 22.71% performance drop ($2.73 \rightarrow 2.11$). In contrast, MotionAgent suffers from a drastic 87.16% performance degradation ($7.79 \rightarrow 1.00$), with MT-Bench scores across all topics collapsing to 1 (rated as "completely unreasonable" by GPT-4). This severe collapse is primarily attributed to overfitting on the newly introduced motion-specific tokens (e.g., Motion_index), resulting in catastrophic forgetting of linguistic knowledge.

In comparison, our approach achieves effective task decoupling and knowledge preservation through the proposed MoE LoRA framework. Using the same foundation model, Gemma-2-2b-it, as MotionAgent, our method demonstrates only a marginal 3.34% degradation ($7.79 \rightarrow 7.53$). This clearly highlights the superior capacity of our framework to retain foundational knowledge while adapting to new modalities and tasks. Moreover, our MoE LoRA framework is model-agnostic and can be seamlessly integrated into the instruction tuning process of various foundation models. When applied to Vicuna-7b-v1.5, the primary foundation model used in our study, fine-tuning with MoE LoRA results in only a 6.10% performance drop ($5.90 \rightarrow 5.54$), further demonstrating the broad applicability and robustness of our framework. Additional qualitative results are provided in Appendix A.1.

## 4.2 Evalution on Text-to-Motion Task

For text-to-motion task, we compare our HMVLM with state-of-the-art methods on HumanML3D dataset. Following prior work [21, 73], we use four evaluation metrics: *R precision*(*Top1-Top3*) and *Multi-modal Distance*(*MM-D*) for text-to-motion retrieval accuracy, *Frechet Inception Distance* (FID) for motion realism, and *Diversity* (Div.) for motion variation.

Table 2: Quantitative results of text-to-motion on the HumanML3D dataset

| Methods | R precision↑ | | | FID.↓ | MM-D.↓ | Div.→ |
| --- | --- | --- | --- | --- | --- | --- |
| | Top-1 | Top-2 | Top-3 | | | |
| GT | 0.511 ±.003 | 0.703 ±.003 | 0.797 ±.002 | 0.002 ±.000 | 2.974 ±.008 | 9.503 ±.065 |
| TM2T [22] | 0.424 ±.003 | 0.618 ±.003 | 0.729 ±.002 | 1.501 ±.017 | 3.467 ±.011 | 8.589 ±.076 |
| T2M [21] | 0.455 ±.003 | 0.636 ±.003 | 0.736 ±.002 | 1.087 ±.021 | 3.347 ±.008 | 9.175 ±.083 |
| MDM [60] | 0.320 ±.005 | 0.498 ±.004 | 0.611 ±.007 | 0.544 ±.044 | 5.566 ±.027 | 9.559 ±.086 |
| MD [74] | 0.491 ±.001 | 0.681 ±.001 | 0.782 ±.001 | 0.630 ±.001 | 3.113 ±.001 | 9.410 ±.049 |
| MLD [6] | 0.481 ±.003 | 0.673 ±.003 | 0.772 ±.002 | 0.473 ±.013 | 3.196 ±.010 | 9.724 ±.082 |
| T2M-GPT [73] | 0.491 ±.003 | 0.680 ±.003 | 0.775 ±.002 | 0.116 ±.004 | 3.118 ±.011 | **9.761** ±.081 |
| ReMoDiffuse [75] | 0.510 ±.005 | 0.698 ±.006 | 0.795 ±.004 | 0.103 ±.004 | 2.974 ±.016 | 9.018 ±.075 |
| AttT2M [79] | 0.499 ±.003 | 0.690 ±.002 | 0.786 ±.002 | 0.112 ±.006 | 3.038 ±.007 | 9.700 ±.090 |
| MoMask [20] | **0.521** ±.002 | **0.713** ±.002 | **0.807** ±.002 | **0.045** ±.002 | **2.958** ±.008 | 9.620 ±.064 |
| MotionGPT [76] | 0.364 ±.005 | 0.533 ±.003 | 0.629 ±.004 | 0.805 ±.002 | 3.914 ±.013 | 9.972 ±.026 |
| MotionGPT [32] | 0.492 ±.003 | 0.681 ±.003 | 0.733 ±.006 | 0.232 ±.008 | 3.096 ±.008 | 9.528 ±.071 |
| MotionAgent [67] | 0.482 ±.004 | 0.672 ±.003 | 0.770 ±.002 | 0.491 ±.019 | 3.138 ±.010 | 9.838 ±.244 |
| MotionGPT-2 [65] | 0.496 ±.002 | 0.691 ±.003 | 0.782 ±.004 | 0.191 ±.004 | 3.080 ±.013 | **9.860** ±.026 |
| Ours (single task) | **0.502** ± .003 | **0.692** ±.004 | **0.785** ±.002 | **0.123** ±.004 | **3.039** ±.027 | 9.443 ±.132 |
| Ours | 0.463 ± .006 | 0.646 ±.004 | 0.744 ±.001 | 0.156 ±.010 | 3.328 ±.004 | 9.544 ±.161 |

Tab. 2 presents quantitative comparisons. Methods in the upper section are task-specific T2M models trained from scratch, whereas methods in the lower section are multimodal frameworks based on foundation language models, fine-tuned via instruction tuning. Among these, MoMask [20] achieves state-of-the-art results in most metrics due to its cascaded mask Transformer design; however, such specialized approaches often lack scalability and generalizability for broader multimodal tasks.

Among multimodal foundation models, we evaluate our proposed HMVLM in two settings (lower section of Tab. 2). *Ours (single task)* denotes evaluation with only T2M task fine-tuning, using the same MoE LoRA architecture. Our model demonstrates remarkable performance across most metrics, thanks to dynamic expert assignment via MoE and fine-grained body-part tokenization (detailed in Sec. 4.5). Under multi-task fine-tuning (*Ours*), HMVLM remains competitive across all metrics, although its performance decreases compared to the single-task setting. This is because, in a single-task setting, all LoRA experts focus solely on one downstream task within the same parameter budget.

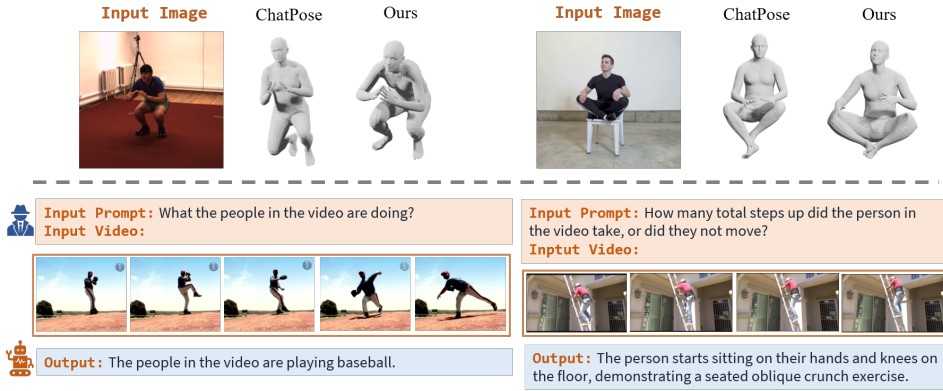

Figure 4: Qualitative results for human pose estimation and human video understanding.

## 4.3 Evaluation on Human Vision Tasks

**Human Pose Estimation.** We follow the evaluation setup of ChatPose [16] using the MPJPE (Mean Per-Joint Position Error) and PA-MPJPE (Procrustes-Aligned MPJPE) as metrics. As shown in Tab. 4, our method surpasses ChatPose—a comparable foundation-model-based approach, in the single-task fine-tuning scenario. This result highlights the advantage of the MoE LoRA architecture in enabling fine-grained expert assignment. Qualitative comparisons with ChatPose, presented in Fig. 4, demonstrate our method's superior accuracy in capturing limb details using examples from the Human3.6M (left) and MoViD (right) datasets, validating the effectiveness of our body-part-based tokenization strategy.

**Motion Video Understanding.** Fig. 4 illustrates an example of our model's performance in human motion video comprehension and reasoning tasks. Our method successfully identifies motion categories (e.g., baseball motion at the top-left) and exhibits spatio-temporal reasoning capabilities. For instance, the bottom-left corner shows the model accurately determining the number of steps climbed, while the right side provides concise descriptions of the character's movements and postures.

## 4.4 Expert Weight Distribution

We analyze the average expert weights from the gating network across different tasks to evaluate the MoE LoRA architecture's task-decoupling capability. Specifically, we extract textual features by inputting each task's instructions and prompts into CLIP's text encoder, then compute the expert weights using the gating network. As shown in Table 3, for the general dialogue task (GD), the loss in Equation 4 encourages the gating network to prioritize the *zero expert*, thereby preserving the pretrained parameters of the foundation model. For other tasks, the gating network dynamically adjusts the expert weight based on the instruction and prompt, reflecting the idea of divide and conquer.

Table 3: Average gating weights across different tasks. GD, T2M, HPE, and HVU denote general dialogue, text-to-motion, human pose estimation, and human video understanding tasks, respectively.

| Task | Zero Expert | Expert 1 | Expert 2 | Expert 3 | Expert 4 |
|------|-------------|----------|----------|----------|----------|
| GD | 0.999 | $2.364 \times 10^{-6}$ | $5.005 \times 10^{-6}$ | $1.357 \times 10^{-6}$ | $1.827 \times 10^{-6}$ |
| T2M | 0.694 | 0.052 | 0.067 | 0.085 | 0.102 |
| HPE | 0.454 | 0.292 | 0.084 | 0.106 | 0.063 |
| HVU | 0.167 | 0.252 | 0.150 | 0.013 | 0.418 |

Table 4: Quantitative results of pose estimation on the H3.6M and 3DPW datasets

| Methods | H3.6M | | 3DPW | |
|---------|-------|-------|------|------|
| | MPJPE↓ | PA-MPJPE↓ | MPJPE↓ | PA-MPJPE↓ |
| SPIN [36] | 61.9 | 42.6 | 102.9 | 62.9 |
| HMR 2.0 [19] | **50.0** | **33.6** | **91.0** | **58.4** |
| ChatPose [16] | 126.0 | 82.4 | 163.6 | 81.9 |
| Ours(single task) | **92.8** | **55.3** | **105.3** | **56.24** |
| Ours | 114.7 | 64.8 | 127.7 | 63.3 |

## 4.5 Ablation Study

HMVLM incorporates the MoE LoRA architecture and a body-part-based tokenization strategy. Accordingly, we conduct ablation studies focusing on these two key components and model efficiency.

**Effectiveness of Body Part-based Tokenization.** To evaluate the spatial modeling capability of our proposed body-part-based tokenization, we conduct ablation studies on both motion reconstruction and T2M performance. As shown in Tab. 5, the baseline tokenizer ("W/o BP") corresponds to the standard whole-body motion tokenizer described in Sec. 3.4. Although enlarging the codebook size

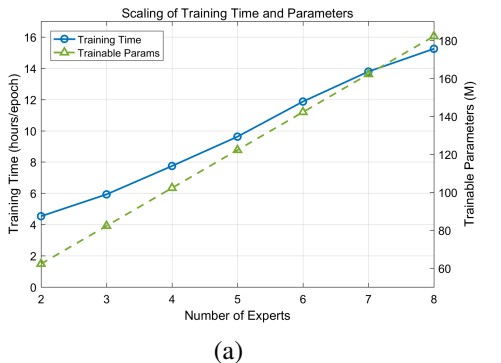
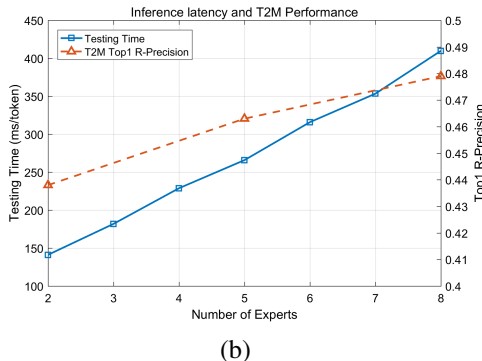

|  (a)  |  (b)  |

Figure 5: Efficiency analysis of the MoE LoRA model under different numbers of experts. (a) Training time and parameter scaling. (b) Inference latency and T2M performance.

(i.e., $K$ × number of body parts) increases the model capacity, using a single codebook for the entire body fails to enhance spatial expressiveness. In contrast, our body-part-based tokenizer yields clear improvements in R-precision and reconstruction MSE, demonstrating its superior spatial modeling ability. The slightly higher FID may result from combining multiple part-specific codebooks, which increases pose diversity but introduces a minor distribution shift.

Table 5: Abalation study on different tokenizers. $K$ represent the codebook size.

| Methods | R precision(Top-3)↑ | FID.↓ | MM-D.↓ | Div.→ | MSE. ↓ |
|---|---|---|---|---|---|
| Ours (single task) | 0.785 ±.002 | 0.123 ±.004 | 3.039 ±.027 | 9.443 ±.132 | 0.966 |
| W/o BP (K=512) | 0.741 ±.001 | 0.336 ±.012 | 3.291±.003 | 9.000 ±.263 | 1.377 |
| W/o BP (K=512*5) | 0.758 ±.003 | 0.110 ±.008 | 3.232 ±.016 | 9.508 ±.212 | 1.34 |

**Effectiveness of $\mathcal{L}_{gat}$.** We investigate the impact of the $\mathcal{L}_{gat}$ on preserving the foundation model's world knowledge. As shown in Tab. 1, removing the $\mathcal{L}_{gat}$ results in catastrophic forgetting, attributed to modifications in pre-trained parameters and changes in the prediction head that make the model overly focussed on newly introduced motion tokens. These findings support the effectiveness of combining the the MoE LoRA architecture with $\mathcal{L}_{gat}$ for building robust multimodal frameworks while mitigating catastrophic forgetting.

**Efficiency of MoE LoRA.** We evaluate the efficiency of the MoE LoRA architecture under different numbers of experts. As shown in Fig. 5(a), the number of trainable parameters and training time increase nearly linearly with the number of experts, as each LoRA expert shares the same rank. For inference latency, as shown in (b), the LoRA weights must be dynamically combined according to the gating network's output, preventing pre-merging with the pretrained model and causing a moderate rise in latency. We also observe the T2M Top-1 R-precision improves but gradually saturates as experts increase. Considering the trade-off between efficiency and performance, we adopt five experts.

## 5 Discussion

In this paper, we present HMVLM, a MoE LoRA-based multimodal framework designed for a range of human-centric tasks, including motion perception, comprehension, and generation. By leveraging the MoE architecture and the introduction of a *zero expert*, our approach preserves the foundation model's world knowledge and generative capabilities during instruction tuning. Furthermore, we incorporate a spatial Transformer to independently encode body part features into discrete tokens, enabling precise and fine-grained motion and pose representations.

**Limitations and Future Work.** While HMVLM advances the joint modeling of human motion, language and vision, several limitations remain. The modality connections are still learned in a independent pairwise manner, limiting holistic integration. Additionally, domain discrepancies across datasets hinder the model's ability to perform seamless any-to-any generation.

## Acknowledgment

This work was supported by National Key R&D Program of China (NO. 2022YFB3303202) and the Innovation Funding of ICT, CAS under Grant NO. E561010

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

# Appendix

This appendix provide qualitative results(Sec.A), additional experiments(Sec.B),visulization(Sec.C) and implementation details(Sec.D).

**Video.** We also provide the supplementary video to showcase our comparisons with SOTA methods and application examples of our approach, including text-to-motion pose estimation, and human motion video understanding.

## A  Qualitative Results

### A.1  Qualitative Comparison of Knowledge Retention

Fig. 6 presents a qualitative comparison of the knowledge retention in foundation models across different methods, including MotionAgent [67], MotionGPT [76], and our method. The input prompt is sampled from the MT-Bench writing topic.

The left side of Fig 6 shows the foundation model outputs before instruction tuning on the text-to-motion task. All of the models generate well-structured, relevant emails with reasonable suggestions, fully satisfying the prompt requirements. The right side of the figure shows the results after instruction tuning. MotionAgent nearly completely loses the dialog capability of its foundation model, with outputs overwhelmed by repetitive "<Motion_index>" tokens. This is caused by the introduction of a new motion vocabulary, modifications to the prediction head, and the application of LoRA matrices across all Transformer modules (Query, Key, Value, and Projection), resulting in severe overfitting to the motion-related task and catastrophic forgetting of the original abilities. MotionGPT, which preserves the original prediction head and applies LoRA only to the Query and Value matrices, retains a portion of the foundation model' knowledge. However, its responses are noticeably more terse, suggesting a certain degree of knowledge degradation. Moreover, Tab. 2 in the main text shows that this fine-tuning strategy cannot achieve superior downstream task performance.

In contrast, our method adopts the MoE LoRA architecture, which enables expert routing through a gating network even when LoRA is applied across all linear modules. The *zero expert* mechanism allows the model to fallback to the original parameter path when handling normal dialogue tasks, effectively preserving the foundational knowledge and achieving a good balance between avoiding model forgetting and enhancing downstream task performance.

### A.2  Qualitative Comparison of Text-to-Motion

We compare our method with state-of-the-art foundation model-based T2M approaches on the HumanML3D test set. We use their publicly released pretrained models and LoRA parameters. The results are shown in Fig 7, poses or motions with semantic errors are highlighted with red boxes. In the left example, the prompt requires a squat followed by running in place, but neither MotionAgent nor MotionGPT generates the squatting motion. More qualitative results can be found in supplementary video.

### A.3  Qualitative Comparison of Pose Estimation

We conducted qualitative comparisons on the 3DPW dataset, following its official training and test split. we compare our method with ChatPose [16], as shown in Fig. 8. The results demonstrate that our approach outperforms ChatPose in body pose accuracy and semantic consistency.

### A.4  Human Motion Video Understanding

**Human motion RGB video understanding.** We provide some examples of human motion video understanding. The results are shown in Fig. 9, all the video are extracted from MoVid dataset [5]. For more qualitative results, please refer to our supplementary video.

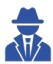

Write a persuasive email to convince your introverted friend, who dislikes public speaking, to volunteer as a guest speaker at a local event......

Subject: Speaking Up for a Cause You Care About
Hey [Friend's Name],

Remember how you were talking about [mention their area of expertise/passion]? I was thinking, wouldn't it be amazing to share your knowledge and passion with the community? I know.......

Best,
[Your Name]

Foundation Model (`Gemma-2-2b-it`)

Write a persuasive email to convince your introverted<Motion_55> to make a presentation at a local event. Use compelling arguments and address potential objections<Motion_458><Motion_358>......<Motion_196><Motion_243><Motion_358>W<Motion_243>W<Motion_247><Motion_89><Motion_243><Motion_247><Motion_89><Motion_128><Motion_247><Motion_128>of<Motion_243><Motion_247><Motion_128>of<Motion_247><Motion_247><Motion_247>.......

After T2M Instruction Tuning (MotionAgent)

It is important to consider your friend's interests and preferences when planning the event. Here are some suggestions to help you craft a compelling email:
1. Make the first paragraph of your email the most compelling one. In this opening paragraph, establish your friend's expertise and convey the significance of the event.
2. Explain why your friend should be a guest speaker. Share relevant information from your friend's background......

Foundation Model (`Llama-2-7b`)

Here is a persuasive email to encourage your friend to volunteer as a guest speaker at an upcoming local event:

1. Share your friend's enthusiasm for the event's cause.
2. Communicate how volunteering at this event will help your friend develop public speaking skills.
3. Use logic to explain how volunteering at this event will help your friend grow professionally......

After T2M Instruction Tuning (MotionGPT)

Subject: Volunteer as a Guest Speaker at a Local Event
Dear [Friend's Name],
I hope this email finds you well. I understand that public speaking is not your forte......
Here are a few compelling arguments to consider:
1. Share your expertise: You have a wealth of knowledge and experience in your field. By sharing your expertise, you can inspire and educate others, making a positive impact on the community......

Foundation Model(`vicuna-7b-v1.5`)

Subject: Let's challenge our comfort zones together!
Dear [Friend's Name],
I hope this email finds you well. I was thinking about the upcoming local event......
Here are a few compelling reasons why you should consider volunteering as a guest speaker:
1. Opportunity to share your expertise: You have a unique perspective and valuable knowledge to share. By speaking at the event, you'll have the chance to educate and inspire others......

After T2M Instruction Tuning(Ours)

Figure 6: Qualitative results of question-answering dialogues from foundation language models with different methods, before and after instruction tuning on text-to-motion tasks.

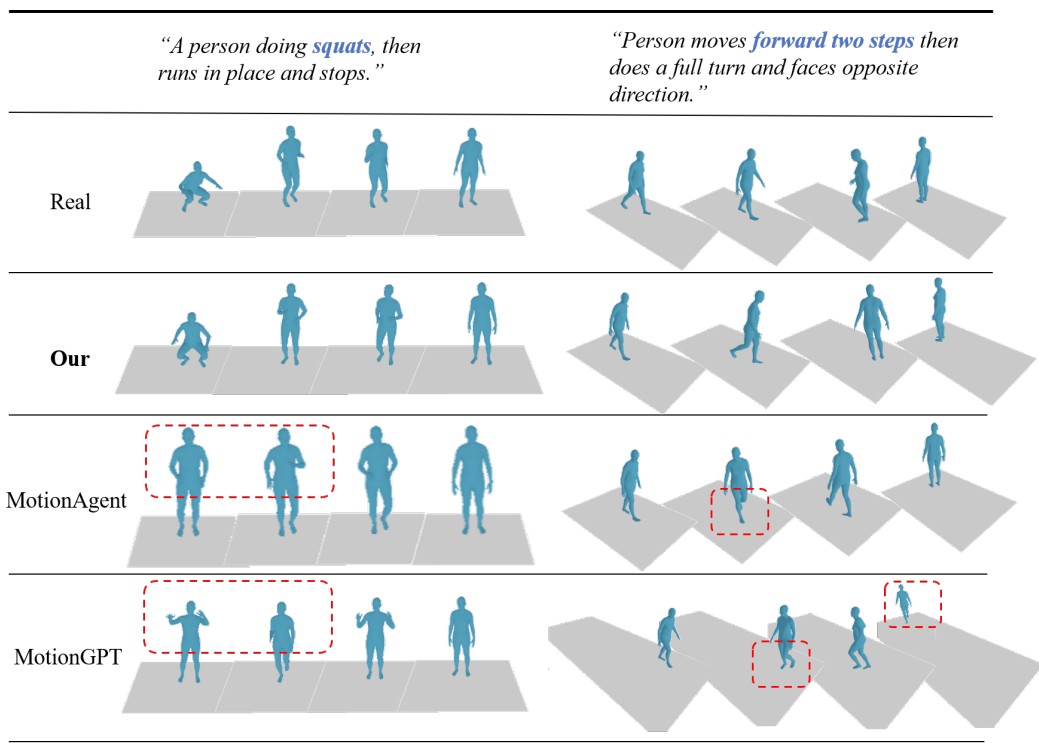

Figure 7: Qualitative comparison on text-to-motion task. The provided state-of-the-art methods are under the same training and inference setting on HumanML3D [21]. The red box highlights the poses that do not match the prompt semantics.

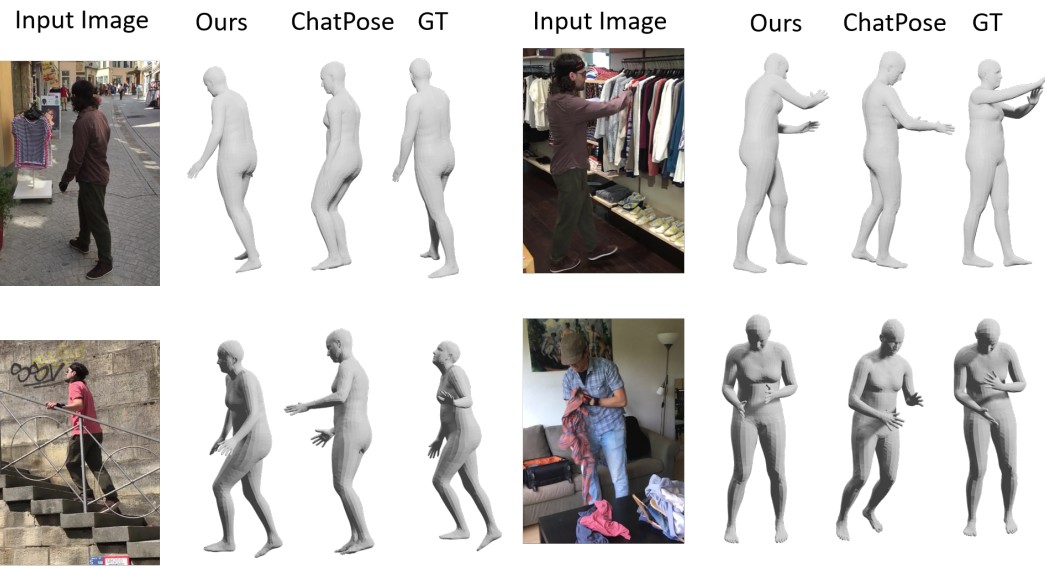

Figure 8: Qualitative comparison with the ChatPose [16] on the 3DPW test set.

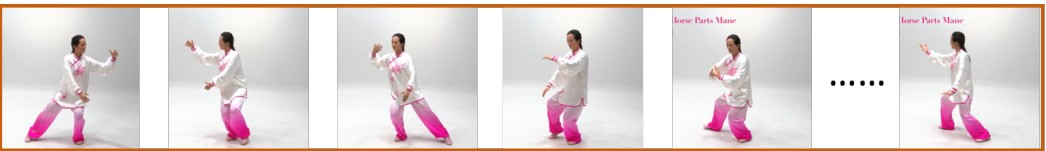

**Input Prompt:** What kind of exercise is the woman in the video doing?

**Output:** The woman in the video is performing the G'an P'ang exercise, a Tai Chi form that involves precise movements of the hands and body while standing in a specific pose.

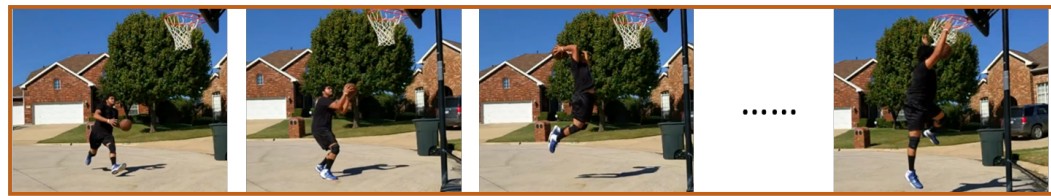

**Input Prompt:** Describe the human motion video concisely.

**Output:** The video shows a man performing a windmill dunk while wearing basketball shoes. He starts with a hand toss of the basketball and ends with a dunk. The camera zooms in and out during the action.

Figure 9: Examples of human motion RGB video understanding.

**Semantic cycle invariance testing.** We combine text-to-motion generation with video understanding tasks to evaluate whether the semantic consistency is preserved after the cycle. As shown in the qualitative results in Fig. 10, we first generate a 3D motion based on the input text prompt, then render the 3D motion sequence into a 2D character video. The model is then tasked with understanding the motion video (using a fixed prompt: "Describe the rendered human motion video concisely"), and finally, we examine the semantic alignment between the input and output texts. In the semantic cycle test, we find that the motion semantics are largely preserved, but errors and hallucinations still occur. Our video understanding strategy involves sampling frames from the video and feeding them into the foundation model. However, if key poses are missed during sampling or represent only a small portion of the entire motion sequence, semantic inaccuracies may arise. For example, in the left part of Figure 10, kneeling is mistakenly identified as squatting.

# B    Additional Experiments

The evaluation results on the KIT-ML dataset [49] are shown in Table 6. This section adopts the same evaluation metrics as used for the HumanML3D dataset. Similarly, Our method with single task instruction tuning outperforms other foundation model-based text-to-motion approaches across most metrics.

# C    Visualization

As shown in Fig. 11, we visualize the forgetting effects on foundation language models before and after the text-to-motion task for our method (Vicuna-7B-v1.5 based), our method (Gemma-2-2B-it based), MotionGPT, and MotionAgent under the MT_Bench [78] benchmark. This corresponds

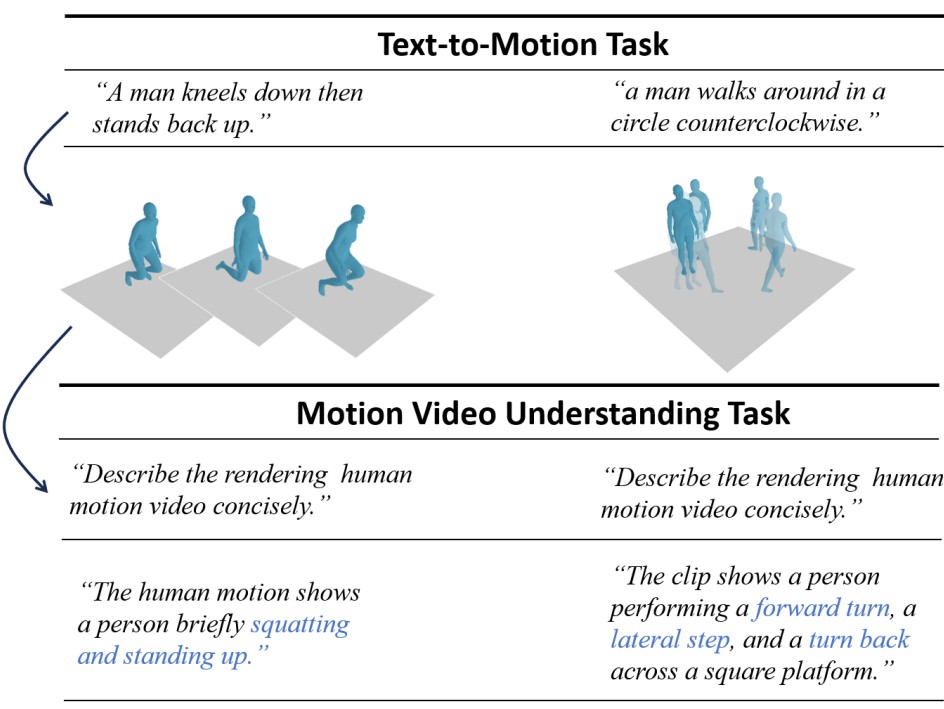

Figure 10: Semantic loop invariance testing.

to Tab. 1 in the main text. MT_Bench covers eight topics, including writing, roleplay, extraction, reasoning, and more. In Fig. 11, greater overlap of the circles indicates less forgetting in the foundation models. It can be observed that our method (based on Gemma-2-2B-it) and MotionAgent exhibit significant differences in their anti-forgetting ability under the same foundation model, which is attributed to our MoE LoRA architecture.

## D    Implementation Details

We conduct experiment on a single NVIDIA A800 80G GPU. In the pose and motion tokenizer, the number of body parts is set to 5, corresponding to the torso and four limbs with each part having an embedding dimension of $S = 512$. The codebook size for each body part is fixed at $K = 512$ and temporal compression ratio is set to $l = 4$ in motion tokenization. During training, the commitment loss coefficient $\lambda_{com}$ is set to 0.02. We use the AdamW optimizer with hyperparameters $[\beta_1, \beta_2] = [0.9, 0.99]$ and a learning rate of $2 \times 10^{-4}$.

During the instruction tuning stage, simultaneous fine-tuning on three human motion-related tasks took a total of 120 hours. We used a batch size of 32 and a micro batch size of 2. AdamW is also used for optimization in this stage, with an initial learning rate of $3 \times 10^{-3}$, which is scheduled using Cosine Annealing.

When calculating the MoE LoRA inference latency (as shown in Figure 5 (b)), we use Vicuna-7b-v1.5 as the foundation model. The inference time is the sum of the gating network computation and multimodal inference time. We conduct latency tests with a batch size of 1 and without using caching. Since the model's inference time is related to the number of input tokens, we fix the input token length to 84 when testing latency under different numbers of experts, as this is the average token length in the T2M test set.

Table 6: Quantitative results of text-to-motion on the KIT-ML dataset

| Methods | R precision↑ | | | FID.↓ | MM-D.↓ | Div.→ |
|---|---|---|---|---|---|---|
| | Top-1 | Top-2 | Top-3 | | | |
| GT | 0.424 ±.005 | 0.649 ±.006 | 0.779 ±.006 | 0.031 ±.004 | 2.788 ±.012 | 11.080 ±.097 |
| TM2T [22] | 0.280 ±.006 | 0.463 ±.007 | 0.587 ±.005 | 3.599 ±.153 | 4.591 ±.026 | 9.473 ±.117 |
| T2M [21] | 0.361 ±.006 | 0.559 ±.007 | 0.681 ±.007 | 3.022 ±.107 | 3.488 ±.028 | 10.720 ±.145 |
| MDM [60] | 0.164 ±.004 | 0.291 ±.004 | 0.396 ±.004 | 0.497 ±.021 | 9.191 ±.022 | 10.847 ±.109 |
| MD [74] | 0.417 ±.004 | 0.621 ±.004 | 0.739 ±.004 | 1.954 ±.062 | 2.958 ±.005 | **11.100** ±.143 |
| MLD [6] | 0.390 ±.008 | 0.609 ±.008 | 0.734 ±.007 | 0.404 ±.027 | 3.204 ±.027 | 10.800 ±.117 |
| T2M-GPT [73] | 0.416 ±.006 | 0.627 ±.006 | 0.745 ±.006 | 0.514 ±.029 | 3.007 ±.023 | 10.921 ±.108 |
| ReMoDiffuse [75] | 0.427 ±.014 | 0.641 ±.004 | 0.765 ±.055 | **0.155** ±.006 | 2.814 ±.012 | 10.800 ±.105 |
| AttT2M [79] | 0.413 ±.006 | 0.632 ±.006 | 0.751 ±.006 | 0.870 ±.039 | 3.039 ±.021 | 10.960 ±.123 |
| MoMask [20] | **0.433** ±.007 | **0.656** ±.005 | **0.781** ±.005 | 0.204 ±.011 | **2.779** ±.022 | 10.711 ±.087 |
| MotionGPT [76] | 0.340 ±.002 | 0.570 ±.003 | 0.660 ±.004 | 0.868 ±.032 | 3.721 ±.018 | 9.972 ±.026 |
| MotionGPT [32] | 0.366 ±.005 | 0.558 ±.004 | 0.680 ±.005 | 0.510 ±.016 | 3.527 ±.021 | 10.350 ±.084 |
| MotionAgent [67] | 0.409 ±.006 | 0.624 ±.007 | 0.750 ±.005 | 0.781 ±.026 | 2.982 ±.022 | **11.407** ±.103 |
| MotionGPT-2 [65] | **0.427** ±.003 | 0.627 ±.002 | 0.764 ±.003 | 0.614 ±.005 | 3.164 ±.013 | 11.256 ±.026 |
| Ours (single task) | 0.423 ±.004 | **0.643** ±.003 | **0.769** ±.004 | **0.306** ±.014 | **2.848** ±.016 | 11.175 ±.093 |
| Ours | 0.381 ± .005 | 0.585 ±.003 | 0.680 ±.013 | 0.567 ± .028 | 3.404 ±.019 | 10.595 ±.125 |

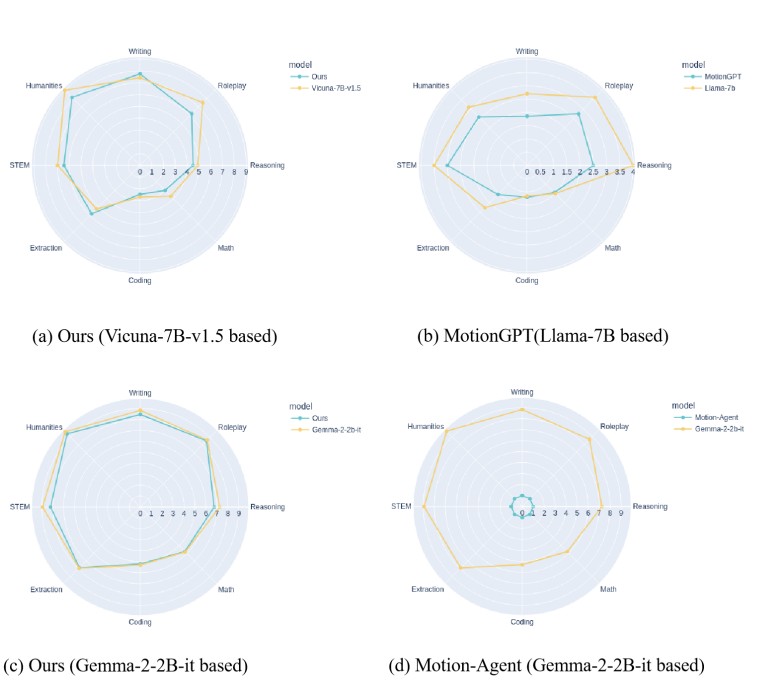

(a) Ours (Vicuna-7B-v1.5 based)  (b) MotionGPT(Llama-7B based)

(c) Ours (Gemma-2-2B-it based)  (d) Motion-Agent (Gemma-2-2B-it based)

Figure 11: Visualization of the forgetting levels before and after T2M fine-tuning across different methods.

## D.1 Evaluation Metrics

**R-precision:** This metric evaluates the consistency between the generated motion and the textual description. Specifically, a generated motion is paired with its corresponding ground-truth text and 31 randomly selected unrelated text descriptions to form a candidate set. Features of the motion and all text descriptions are extracted using their respective encoders [21], and pairwise feature distances are computed. These distances are then ranked in ascending order. The probabilities of the ground-truth text appearing in the Top-1, Top-2, and Top-3 positions are reported as the evaluation result. Higher values indicate that the generated motion better aligns with the semantic description.

**FID:** This metric measures the distributional difference between generated motions and real motions from the dataset. A lower FID score indicates that the generated motions are more similar to real samples in terms of overall feature distribution.

**MultiModal Distance (MM-D.):** This metric evaluates semantic alignment by computing the Euclidean distance between the text feature and the corresponding generated motion feature. A smaller value indicates better semantic matching.

**Diversity (Div):** This metric assesses the diversity of motions generated by the model. Specifically, two subsets, each containing 300 randomly selected generated motions, are sampled. The average Euclidean feature distance between the two subsets is calculated. A larger value indicates greater diversity in the generated results.

**MPJPE:** Mean Per Joint Position Error measures the average Euclidean distance between the predicted and ground-truth joint positions of a generated 3D human pose. It is computed by calculating the Euclidean distance for each joint in every frame and then averaging over all joints and frames. This metric reflects the spatial accuracy of the generated motion, with lower values indicating closer alignment to the ground-truth poses.

**PA-MPJPE:** Procrustes Aligned MPJPE calculates the MPJPE after applying a rigid alignment (including rotation, scaling, and translation) to the predicted poses to remove global transformation differences. This metric focuses on the structural correctness of the predicted pose regardless of its absolute position and scale. Lower values indicate better structural alignment with the ground truth.

### D.2 Instruction Templates

Tab. 7 presents the templates used for our task instruction tuning, including text-to-motion, pose estimation and video understanding. The <Motion_Placeholder>, <Image_Placeholder>,<Video_Placeholder>, and <Caption_Placeholder> respectively represent the motion sequence, image input, video input and textual description from the trianing datasets.

Table 7: Examples of instruction templates for each task

| Task | Input | Output |
|------|-------|--------|
| Text-to-Motion | Generate a sequence of motion tokens matching the following human motion description. | <Motion_Placeholder> |
| | Generate a sequence of motion tokens matching the following human motion description given the initial token <Motion_Placeholder>. | <Motion_Placeholder> |
| | Generate a sequence of motion tokens matching the following human motion description given the last token <Motion_Placeholder>. | <Motion_Placeholder> |
| Pose Estimation | Can you predict the SMPL pose of the person in this image <Image_Placeholder>. | <Pose_Placeholder> |
| | There is a person in the middle of the image, please output this person's SMPL pose <Image_Placeholder>. | <Pose_Placeholder> |
| | What is the human pose in this image? Please respond with SMPL pose <Image_Placeholder>. | <Pose_Placeholder> |
| | What is the person doing in this image? Please output SMPL pose <Image_Placeholder>. | <Pose_Placeholder> |
| | There is a person in the middle of the image, use SMPL to describe the pose <Image_Placeholder>. | <Pose_Placeholder> |
| Video Understanding | Write a terse but informative summary of the following human motion video clip <Video_Placeholder>. | <Caption_Placeholder> |
| | Describe the following human motion video concisely <Video_Placeholder>. | <Caption_Placeholder> |
| | Render a clear and concise summary of the human motion video below <Video_Placeholder>. | <Caption_Placeholder> |
| | Share a concise interpretation of the human motion video provided <Video_Placeholder>. | <Caption_Placeholder> |
| | Relay a brief, clear account of the human motion video shown <Video_Placeholder>. | <Caption_Placeholder> |
| | ... | <Caption_Placeholder> |

