# OpenReview forum: "HMVLM:Human Motion-Vision-Language Model via MoE LoRA"
_NeurIPS.cc/2025/Conference — NeurIPS 2025 poster_

### Official Review · Reviewer_t2iJ · 2025-06-16

**Clarity:** 3
**Significance:** 3
**Originality:** 3
**Rating:** 4
**Confidence:** 4

**Summary:**

This paper proposes the Human Motion-Vision-Language Model (HMVLM), a unified framework that integrates 3D human motion with foundation language models by leveraging the MoE LoRA strategy. HMVLM introduces a ``zero expert’’ in MoE LoRA to preserve pre-trained linguistic knowledge and adopts body-part-specific tokenization to enhance the spatial resolution in motion/pose encoding. The experimental results demonstrate the effectiveness of the proposed framework.

**Questions:**

1.Would further subdividing body parts (e.g., into smaller joint groups) improve representation accuracy?

2.Are the other methods in Table 2 (such as MotionGPT) also fine-tuned for single task?

3.Does the MoE strategy incur greater memory and time overhead? It would be better to add a table for illustration.

**Ethical Concerns:**

["NO or VERY MINOR ethics concerns only"]

**Final Justification:**

I appreciate the authors' efforts to address my concerns. I will maintain my current rating and hope the authors can make the modifications as promised in future versions.

**Limitations:**

yes

**Quality:**

3

**Strengths And Weaknesses:**

**Strengths**:
1. The MoE LoRA framework with a novel zero expert effectively alleviates catastrophic forgetting problem, which is verified by the maintenance of performance in language tasks.
2. Body-part tokenization improves spatial detail in pose encoding, boosting downstream task performance.
3. HMVLM demonstrates competitive results across diverse tasks under multi-task fine-tuning, highlighting its generalizability.

**Weaknesses**:
1. Some of the symbolic expressions in the paper are difficult for readers to understand. It would be better to simplify them appropriately.
2. Some relevant works are missing. For example, [1] also proposed a pre-trained model for human motion; [2] also addressed the problem of catastrophic forgetting in human motion. These could be briefly discussed in related work.

**References**:

[1] Gao Y, Luan P C, Alahi A. Multi-Transmotion: Pre-trained Model for Human Motion Prediction[J]. arXiv preprint arXiv:2411.02673, 2024.

[2] Tang J, Sun J, Lin X, et al. Temporal continual learning with prior compensation for human motion prediction[J]. Advances in Neural Information Processing Systems, 2023, 36: 65837-65849.

---

> ### Author Rebuttal · Authors · 2025-07-30
>
> ### We sincerely thank the reviewer for the constructive feedback and the time spent evaluating our work. We also appreciate the recognition of the novelty of our method. Point-by-point responses were provided with the aim of addressing the reviewer’s concerns.
>
> Q：Some of the symbolic expressions in the paper are difficult for readers to understand. It would be better to simplify them appropriately.
>
> -  We will simplify and reorganize the symbols in the main text.
>
>
> Q: Some relevant works are missing.
>
> -  We will add the suggested citations in the appropriate sections of the related work and briefly discuss their connections to our approach.
>
>
> Q：Would further subdividing body parts (e.g., into smaller joint groups) improve representation accuracy?
>
> - Finer segmentation (e.g., into smaller joint groups ) could enhance motion modeling precision and decrease the MSE in motion reconstruction. However most current 3D human motion datasets describe motions at the limb-segment level, which aligns well with our chosen granularity.
>
>
> Q: Are the other methods in Table 2 (such as MotionGPT) also fine-tuned for single task?
>
> - Yes. These works apply foundation language models to a single downstream task, i.e., text-to-motion and are therefore fine-tuned for single-task settings.
>
>
> Q: Does the MoE strategy incur greater memory and time overhead? It would be better to add a table for illustration.
>
> - During training, due to the mixture of multiple LoRA experts with the pretrained parameters (see Equation (1)), both the number of trainable parameters and the training time increase linearly with the number of experts. However, during inference, only the computation of the gating network's mixture weights is added. As a result, the increase in inference time is constant and almost negligible. We will add a table for illustration.

---

> ### Comment · Reviewer_t2iJ · 2025-08-05
> **Official Comment by Reviewer t2iJ**
>
> Thank you for the detailed response. I appreciate the authors' effort to address my concerns. Will keep my current rating and hope the authors can make the modifications as promised in future versions.

---

### Official Review · Reviewer_sc8H · 2025-07-01

**Clarity:** 3
**Significance:** 3
**Originality:** 3
**Rating:** 4
**Confidence:** 3

**Summary:**

This paper proposes HMVLM, a unified framework to integrate 3D human motion with vision and language modalities into a foundation language model. The work tackles two critical challenges: the catastrophic forgetting of the model's general knowledge when fine-tuned on specialized motion data, and the difficulty of creating motion representations that are both spatially detailed and compatible with autoregressive models. The authors introduce a MoE LoRA strategy, which notably includes a non-trainable zero expert designed to preserve the foundation model's original weights for non-motion tasks. A gating network dynamically routes inputs to different LoRA experts, guided by a specific loss function to prevent knowledge degradation. They propose a body-part-based tokenizer that partitions the human skeleton into semantic parts and encodes them separately to enhance the spatial resolution of the pose representation. The authors conduct experiments on tasks including text-to-motion generation, pose estimation, and motion video understanding, to show HMVLM's knowledge preservation ability and performance on downstream tasks.

**Questions:**

1. The gating network and the zero expert are central to your contribution. Could you provide more details about the structure of gating network and analysis of the expert weights assigned by the gating network? e.g., showing how the weight for the zero expert changes between linguistic prompts, simple motion tasks, and complex motion tasks would provide excellent insight into whether the framework is behaving as intended.

2. The results in table 2 and 3 show a performance gap between the single task and multi-task models. Could you elaborate on this trade-off? Is this degradation an inherent cost of building a generalized multi-task model with this framework, or potentially mitigated through different training strategies, a different number of experts, or adjustments to the loss functions?

3. The paper mentions partitioning the body into N parts, and Figure 3 seems to suggest 5 parts. How was this specific partition chosen, and how sensitive is the model's performance in both reconstruction and downstream tasks to this design choice, e.g., using more granular parts like hands/feet, or fewer parts.

4. Section 4.1 mentions gating network is trained on LMSYS-Chat-1M. For clarity, could you explicitly state in the main text which datasets were used for the pre-training of the motion and pose tokenizers, respectively? While it can be inferred from Section 3.3 and 4.1, a direct statement would improve readability.

**Ethical Concerns:**

["NO or VERY MINOR ethics concerns only"]

**Final Justification:**

The rebuttal has addressed most of my concerns and strengthened the paper, and I will maintain my score of 4.

**Limitations:**

yes

**Quality:**

4

**Strengths And Weaknesses:**

Strengths:

1. The paper addresses a crucial problem in multimodal AI that how to incorporate specialized modality like human motion in foundation model without it catastrophic forgetting its pre-existing world knowledge.

2. The proposed MoE LoRA framework and zero expert is a novel and simple solution to catastrophic forgetting. The idea of having a non-trainable expert that acts as identity path for the original model combined with a gating mechanism that learns to route general queries to it, is highly intuitive. The experimental results in Table 1, showing only a ~6% performance drop on MT-Bench compared to competitors' >22% or even >87% drops, provide strong evidence for this claim. Crucial ablation studies are provided for the gating loss.

3. Proposed body-part-based tokenizer inspired by patch-based methods in vision is a well-motivated,  which improves the spatial granularity of pose representations.

Weaknesses:

1. The gating network is central to the MoE LoRA framework. However, the paper only provides an ablation study on its training loss, but does not offer any qualitative analysis of the expert weights it learns. It would be insightful to see how these weights are distributed for different types of prompts The architecture of the gating network is not specified as well.

2. In section 4.3, the paper could benefit from a deeper analysis of the performance trade-off observed when moving from single-task to multi-task training. The results in Tables 2 and 3 show a noticeable performance drop in T2M and pose estimation metrics in the multi-task setting compared to the single-task one.

3. The comparison of knowledge preservation in Table 1 is weakened by the fact that the baseline models MotionAgent and MotionGPT use different foundation models than the proposed HMVLM. The authors fairly acknowledge this and focus on the relative performance degradation, but a more direct comparison against a baseline fine-tuned on the same base model would have been more powerful.

---

> ### Author Rebuttal · Authors · 2025-07-30
>
> ### We sincerely thank the reviewer for the constructive feedback and the time spent evaluating our work. We are glad to see the recognition of our contributions and method novelty. In this response, we address the questions point-by-point and aim to resolve the reviewer’s concerns.
>
>
> Q： The paper lacks qualitative analysis and architectural details of the gating network, including how expert weights are distributed for different prompts.
>
> - For the dialogue task, the average gating weights are `[0.999973, 2.3638 * 10^{-6}, 5.0053 * 10^{-6}, 1.3566 * 10^{-6}, 1.8274 * 10^{-6}]`, indicating that the zero expert is predominantly activated to preserve the foundation model’s language capabilities. In contrast, for text-to-motion generation and pose estimation tasks, the average weights are `[0.6938, 0.0522, 0.0671, 0.0850, 0.1018]` and `[0.4543, 0.2919, 0.0844, 0.1062, 0.0632]`, respectively, showing that the model effectively leverages the newly learned LoRA experts for these tasks. The gating network architecture is specified in Appendix D.
>
>
> Q: What causes the performance gap between single-task and multi-task models, and can it be mitigated through training strategies, expert numbers, or loss adjustments?
>
> - The higher performance of single-task training arises because all LoRA expert can focus solely on one downstream task under the same parameter budget.  Multi-task performance can be improved by increasing the number of experts—for example, using 8 experts improves Top-1 R precision in the T2M task by `3.54%`. However, this comes at the cost of higher training complexity due to the expanded MoE configuration. We will add a table in main text for illustration.
>
>
> Q: A fairer comparison would use baselines fine-tuned on the same foundation model as HMVLM.
>
> - We will replace the base model in our ablation to match the one used in the baselines, ensuring a more direct and fair comparison. And it is worth noting that the capability of our MoE LoRA framework to preserve the foundation model’s knowledge is not dependent on the specific choice of foundation model.
>
>
> Q:   How was the 5-part body partition chosen, and how sensitive is performance to using more or fewer parts?
>
> - We follow the body partitioning strategy from [26] and [76] in the main paper reference, which has proven effective in motion modeling and text-to-motion generation. While finer segmentation (e.g., feet and fingers) could enhance motion modeling precision and reconstruction, most current 3D human motion datasets describe motions at the limb-segment level, which aligns well with our chosen granularity.
>
>
> Q: Clarify in the main text which datasets were used to pre-train the motion and pose tokenizers.
>
> - We will explicitly add this clarification in the main text. The motion tokenizer is pre-trained on the HumanML3D training set, while the pose tokenizer is pre-trained on the Human3.6M training set.

---

> > ### Comment · Reviewer_sc8H · 2025-08-05
> >
> > Thank you for your detailed and timely rebuttal. Your responses have effectively addressed my primary concerns.
> >
> > On Q1, the quantitative analysis of the gating network weights provides excellent insight and strongly supports your claims.
> >
> > On Q2, the explanation for the performance trade-off is logical, and the proposed mitigation strategy is convincing.
> >
> > On Q3, I appreciate your commitment to ensuring a fairer comparison in the final version.
> >
> > On Q4 and Q5, the justifications for the body partition choice and the clarification on the tokenizer datasets are both clear and satisfactory.
> >
> > The rebuttal has strengthened the paper, and I will maintain my score of 4.

---

### Official Review · Reviewer_rpyN · 2025-07-01

**Clarity:** 2
**Significance:** 2
**Originality:** 2
**Rating:** 4
**Confidence:** 3

**Summary:**

This paper introduces HMVLM, a unified Human Motion-Vision-Language Model built on a Mixture of Expert Low-Rank Adaptation (MoE LoRA) framework. It targets multiple human-centric tasks, including text-to-motion generation, pose estimation, and motion video understanding. The core techniques are (1) leveraging a gating network to dynamically allocate LoRA expert weights, including a zero-expert to mitigate catastrophic forgetting due to the modality gap between human motion and text, and (2) designing a body-part-based tokenization strategy for fine-grained pose/motion representation compatible with autoregressive architectures. Experimental results show that HMVLM achieves competitive performance on text-to-motion and pose estimation tasks while preserving foundational world knowledge in language models.

**Questions:**

1. The multitask performance appears degraded compared to single-task baselines, and it is unclear why unifying these tasks is practically better. It would be better to give more evidence or analysis on why a unified framework is preferable, for example through cross-task transfer benefits, data efficiency, or user-facing advantages.
2.  In Table 4, the part-based tokenizer does not always outperform a single codebook tokenizer (e.g., worse FID). It would be better to explain why these trade-offs happen, and clarify its benefits on the pose estimation task specifically. Additional analysis or an ablation focused on pose estimation could clarify its usefulness.
3. There is no analysis of the $\alpha_i$ weights across tasks. For instance, what is the average or distribution of $\alpha_0$ on dialogue tasks? Providing such analysis could help validate the effectiveness of the gating mechanism and would strengthen the evidence for the proposed zero expert.
4. It would be better to discuss why simply forcing all $\alpha_i$ to zero (effectively disabling experts on dialogue tasks) would not be a sufficient alternative. A comparison or reasoning about this hard-routing baseline could help me better judge whether the zero expert mechanism is truly necessary.

**Ethical Concerns:**

["NO or VERY MINOR ethics concerns only"]

**Final Justification:**

Thank you for the authors' feedback. The authors have addressed my concerns. I raise my rating from 3 to 4 (Borderline accept).

**Limitations:**

yes

**Quality:**

2

**Strengths And Weaknesses:**

Strengths:
1. This paper uses a technically reasonable framework with a zero expert in MoE to protect pre-trained capabilities.
2. This paper proposes to use part-level tokenisation to improve the presentation.

Weaknesses:
1. The paper is somewhat difficult to understand. The writing is dense and overloaded with technical jargon, making it challenging to follow the core ideas, the architecture details, and the implementation steps. This hinders accessibility. For example, only after careful and repeated reading does it become apparent that the authors intend to describe the model structure first, followed by the training of the tokenizer and then instruction tuning. However, Sec 3.1 and 3.2 actually are parts of 3.4.
2. The authors claim to present a unified framework; however, the experimental results show performance degradation when training on multiple tasks compared to specialized single-task baselines. The paper does not sufficiently justify why combining these tasks in one framework is beneficial in practical or real-world settings.
3. As shown in Table 4, the part-based tokenizer does not consistently outperform a single codebook tokenizer across all metrics (for example, FID is 0.123 vs. 0.110). Some explanation or analysis is missing to justify these trade-offs. Additionally, the impact of the part-based tokenizer on the pose estimation task is not clearly discussed.
4. The paper lacks analysis of the learned gating weights $\alpha_i$ for each task. For instance, it would be helpful to report the average value of $\alpha_0$ for the dialogue task, to better understand how often/much the zero expert is actually selected.
5.  It is unclear whether explicitly setting all $\alpha_i$ values to zero (i.e., bypassing all experts) whenever a task instruction indicates a dialogue task could be a simpler alternative to preserve pre-trained knowledge. A discussion comparing this hard routing strategy to the proposed zero expert would strengthen the justification for its necessity.

---

> ### Author Rebuttal · Authors · 2025-07-30
>
> ### We sincerely thank the reviewer for the constructive suggestions and the time spent our work. We appreciate the recognition of our contributions, including the zero expert in MoE to preserve pre-trained capabilities, and the introduction of part-level tokenization to enhance motion representation. We provide point-by-point responses with the aim of addressing the reviewer’s concerns.
>
>
> Q： The paper is somewhat difficult to understand.
>
> - Our writing aimed to emphasize the core concept of MoE LoRA, starting from the generalized framework (Sec. 3.1) to the details of downstream task fine-tuning (Sec. 3.3, 3.4). We will refine the presentation in the methods section to make the narrative clearer and the core ideas easier to follow.
>
>
> Q: Performance degradation when training on multiple tasks. Why combining different tasks in one framework is beneficial in practical or real-world settings.
>
> - The higher performance of single-task training arises because all LoRA expert can focus solely on one downstream task under the same parameter budget. The MoE LoRA framework integrates multiple tasks into a unified model, enabling automatic selection of the optimal combination of LoRA experts based on the input semantics, without requiring manual task-specific parameter selection.
>
>
> Q: The part-based tokenizer does not outperform the single-codebook tokenizer across all metrics, and its impact on pose estimation is unclear.
>
> - The slightly higher FID of the part-based tokenizer may stem from the combination of multiple part-specific codebooks, which enriches generated pose diversity but introduces a minor distribution shift from the training set. Nevertheless, our method shows clear improvements over the single-codebook tokenizer in key metrics such as R-precision and MSE. We will add the impact of the part-based tokenizer on the pose estimation task.
>
>
> Q: The paper lacks analysis of gating weights, particularly how often the zero expert is selected for each task.
>
> - For the dialogue task, the average gating weights are `[0.999973, 2.3638 * 10^{-6},  5.0053 * 10^{-6}, 1.3566 * 10^{-6}, 1.8274 * 10^{-6}]`, indicating that the zero expert is predominantly activated to preserve the foundation model’s language capabilities. In contrast, for text-to-motion generation and pose estimation tasks, the average weights are `[0.6938, 0.0522, 0.0671, 0.0850, 0.1018]` and `[0.4543,  0.2919, 0.0844, 0.1062, 0.0632]`, respectively, showing that the model effectively leverages the newly learned LoRA experts for these tasks.
>
>
> Q: Could setting all expert weights to zero for dialogue tasks be a simpler alternative to the zero expert?
>
> - The gating network enables automatic routing among experts, allowing the model to switch between downstream tasks without manual selection of LoRA experts. Setting all expert weights to zero can indeed replicate the effect of $\alpha_0 = 1$, but it still requires a classifier-like mechanism to identify dialogue-only inputs. Thus, this routing strategy is not simpler than the proposed zero expert approach, while also lacking its flexibility for mixed or ambiguous instructions.

---

> > ### Author Response · Authors · 2025-08-05
> > **Dear  Reviewer rpyN**
> >
> > Dear Reviewer rpyN,
> >
> > We sincerely thank you for your thoughtful comments on our submission. If you have any further feedback or questions, we would greatly appreciate your continued engagement in the discussion. lf you feel that our responses adequately address your concerns, we kindly hope you consider improving the score accordingly.Thank you!
> >
> > Sincerely,
> > The Authors

---

> > ### Comment · Reviewer_rpyN · 2025-08-06
> > **Follow-up questions**
> >
> > Thank you for the authors' response.
> >
> > After reviewing the rebuttal, I have a follow-up question regarding Q4 and Q5. The authors acknowledge that "setting all expert weights to zero can indeed replicate the effect of $\alpha_0 = 1$," and the response in A4 indicates that the value 0.999973 is very close to 1. This suggests that, rather than requiring a classifier-like mechanism to identify dialogue-only inputs, a simpler approach, such as an if-else condition to distinguish between dialogue-only and multimodal (image/video) inputs, could suffice.
> >
> > Therefore, I remain unconvinced about the necessity of the zero expert and would appreciate further clarification on its role and benefit over simpler alternatives.

---

> ### Author Response · Authors · 2025-08-07
> **Further clarification**
>
> Dear Reviewer rpyN,
>
> Thanks for your feedback! Here is our further clarification regarding zero expert.
>
>
> - Simple approaches, such as if-else conditions, are not applicable to our multimodal scenario.This is because, in addition to multimodal (image/video) inputs, there are also cases involving pure text inputs and other modal outputs. Text-to-motion is one such example, where the input is pure text and cannot be distinguished from dialogue-only tasks by simple rule-based methods. Furthermore, the MoE architecture is not only beneficial for classifying multimodal tasks, but it can also improve the performance of single-modal. For instance, MotionVAE [1] improves performance by using a MoE network to classify different motion categories (walking, running, etc.); MixLORA [2] enhances performance for different dialogue topics through MoE.
>
>
> - Zero expert is necessary for dynamically adjusting the foundation model's weights. The theoretical basis is as follows (using num_expert=2 as an example): $AB =  \alpha_0 * A_0B_0 + (1-\alpha_0)  * A_1B_1 $. The existence of zero expert not only enables the model to mix different LoRA matrices at varying proportions but also adjust the absolute values of AB. When the value of $\alpha_0$ is large, the model tends to favor the original model's knowledge,  while the MotionGPT or MotionAgent method can only rely on the fixed AB and fixed merged weights. This flexibility enables our model to outperform other regular LoRA-based methods in multiple downstream tasks (see Tables 2 and 3 in the main text).
>
> -  Overall, zero expert can avoid catastrophic forgetting during multimodal model fine-tuning, while improving task performance by dynamically adjusting mixture weights and scale.
> ```
> [1] Ling, Hung Yu, et al. "Character controllers using motion vaes." ACM Transactions on Graphics (TOG) 39.4 (2020): 40-1
>
> [2] Li D, Ma Y, Wang N, et al. Mixlora: Enhancing large language models fine-tuning with lora-based mixture of experts [J]. CoRR, 2024.
> ```

---

> > ### Comment · Reviewer_rpyN · 2025-08-08
> > **My follow-up question has been addressed.**
> >
> > Thank you for the authors' feedback. The authors have addressed my follow-up concerns via good explanations and an example. I will raise my rating from 3 to 4 (Borderline accept).

---

### Official Review · Reviewer_C5Mg · 2025-07-05

**Clarity:** 3
**Significance:** 2
**Originality:** 2
**Rating:** 3
**Confidence:** 4

**Summary:**

This paper presents HMVLM, a unified multimodal framework that integrates 3D human motion with foundation language models. It focuses on two core challenges: mitigating catastrophic forgetting during multimodal instruction-tuning and designing generalizable, autoregressive-compatible pose representations. To address these, the authors propose a Mixture of Expert Low-Rank Adaptation (MoE LoRA) approach, incorporating a "zero expert" to preserve general linguistic capabilities of the foundation model. Additionally, they introduce a body-part-specific tokenization scheme to enhance spatial granularity in motion representation. The framework is evaluated on multiple tasks, including text-to-motion generation, monocular pose estimation, and motion video understanding, showing promising performance and improved resistance to knowledge forgetting.

**Questions:**

1. Zero Expert Selection: Could the authors clarify how the gating network decides when to activate the zero expert? Is there a learned threshold or attention-based mechanism involved?

2. Efficiency of MoE LoRA: While LoRA is known for being parameter-efficient, how does the proposed MoE variant affect training and inference cost, especially as the number of experts increases?

3. Extending to Other Modalities: Could this framework be adapted to include other modalities such as audio or haptics? If so, what are the expected challenges?

4. Sensitivity to Body Part Definitions: How sensitive is performance to the way the body is partitioned? Is there a principled way the authors chose the granularity of the segmentation?

5. Comparison to Other MoE Architectures: How does this method compare to existing MoE models in terms of scalability, performance, and interpretability?

**Ethical Concerns:**

["NO or VERY MINOR ethics concerns only"]

**Limitations:**

- Modality interactions are modeled in a pairwise fashion, which may not fully exploit cross-modal synergies.

- Generalization to new domains is limited due to dataset-specific training and domain shifts.

- The paper lacks detailed ablations on expert configuration and gating behavior.

- Qualitative results are underrepresented in the main text, making it harder to fully assess the model’s output quality.

**Quality:**

3

**Strengths And Weaknesses:**

**Strengths**

- Mitigation of Catastrophic Forgetting: The use of a "zero expert" within the MoE LoRA framework is a compelling idea. It allows the model to retain general language capabilities while adapting to new motion-related tasks, and empirical results support its effectiveness (e.g., only 6.07% drop vs. larger drops in baselines).

- Unified Framework for Multiple Tasks: HMVLM supports various human-centric tasks simultaneously, marking a step forward compared to existing task-specific pipelines.

- Fine-Grained Pose Representation: The proposed body-part-aware tokenization leads to more granular motion encoding, which appears to benefit downstream performance. The approach draws a useful analogy to patch-based image tokenization and applies it in a novel context.

- Strong Experimental Results: Across several benchmarks, the model achieves competitive or superior performance, particularly in text-to-motion generation (e.g., HumanML3D). These results highlight the potential of the proposed architecture.

**Weaknesses**

- Limited Qualitative Analysis: While quantitative results are solid, the paper could be strengthened by including more qualitative visualizations, especially for complex tasks like motion generation. The supplementary materials are mentioned but not integrated into the main paper, limiting the reader’s ability to assess this aspect.

- Modality Fusion Strategy: The framework currently models modality interactions in a pairwise manner. This may limit the depth of multimodal integration, especially when extending to more complex scenarios requiring holistic reasoning across all modalities.

- Domain Generalization: The model’s performance across diverse datasets is affected by domain discrepancies, which challenges its generalization to unseen settings and limits its flexibility for open-world deployment.

- Ablation on MoE Configuration: There is limited discussion on how the number of experts or their specific configurations affect the model. An ablation study would provide deeper insights into the trade-offs involved.

---

> ### Author Rebuttal · Authors · 2025-07-30
>
> ### We sincerely thank the reviewer for constructive feedback and the time spent evaluating our work. We appreciate the recognition of our key contributions, including zero expert for catastrophic forgetting mitigation, strong results and so on. We provide point-by-point responses with the aim of resolving the reviewer’s concerns.
>
> Q: Limited Qualitative Analysis
>
> - Given the page limit, we will rearrange qualitative results in the main paper, highlighting motion generation, with additional examples in the appendix and supplementary video.
>
>
> Q: Modality Fusion Strategy
>
> - While our current framework adopts supervised pairwise training for several downstream tasks, this approach enables the foundation model to effectively absorb knowledge from other modalities and achieve strong performance across different tasks. Additionally, modality fusion in our system is primarily mediated by text, for example, as demonstrated in the supplementary video, we illustrate a loop from text-driven motion generation to 3D motion understanding. This demonstrates that our model is capable of learning generalizable fusion knowledge from different pairwise datasets to a certain extent.
>
>
> Q: Domian Generalization
>
> - Domain discrepancies are inevitable in multimodal learning. As discussed in the manuscript, the large gap between 3D human motion and text often leads to catastrophic forgetting when introducing and fine-tuning 3D motion data. Our proposed MoE LoRA framework with zero expert effectively mitigates domain discrepancies across datasets. Further improving generalization to unseen open-world scenarios would require scaling up data, which is beyond the scope of our work and contribution.
>
>
> Q: Ablation on MoE Configuration
>
> - Increasing the number of MoE experts allows for more flexible combinations and can improve task performance; however, it also leads to a linear increase in training cost. In our implementation, since our method currently supports three downstream tasks and requires a dedicated "zero expert" as well as an additional expert for knowledge sharing, we set the number of MoE LoRA experts to 5. We also have conducted an ablation study on the number of experts. When the number of experts is set to 2 (including the zero expert), the performance on the T2M and Pose Estimation tasks decreased by `5.40 %` and `15.9%`, repectively. Conversely, increasing the number of experts to 8 resulted in performance improvements of `3.54%` and `9.76%` on the T2M and Pose Estimation. We will add a table in main text for illustration.
>
>
> Q: Zero Expert Selection
>
> - As illustrated in the lower-left part of Figure 2, the input text features are encoded and fed into the gating network, which performs a classification-like operation. Owing to the design of loss function (7), we explicitly encourage the gating network to produce a weight distribution that is close to $\alpha_0 =1$ for pure text-based dialogue tasks, thereby preferentially activating the zero expert.
>
>
> Q: Efficiency of MoE LoRA
>
> - During training, due to the mixture of multiple LoRA experts with the pretrained parameters (see Equation (1)), both the number of trainable parameters and the training time increase linearly with the number of experts. However, during inference, only the computation of the gating network's mixture weights is added. As a result, the increase in inference time is constant and almost negligible.
>
>
> Q: Extending to Other Modalities
>
> - The proposed framework can be extended to other modalities. However, this work primarily focuses on a multimodal framework related to 3D human motion and addresses the challenge of catastrophic forgetting observed in previous works such as MotionGPT and MotionAgent.
>
>
> Q：Sensitivity to Body Part Definitions
>
> - We follow the body partitioning strategy from [26] and [76] in the main paper reference, which has proven effective in motion modeling and text-to-motion generation. While finer segmentation (e.g., hands and fingers) could enhance motion modeling precision, most current 3D human motion datasets describe motions at the limb-segment level, which aligns well with our chosen granularity.
>
>
> Q:  Comparison to Other MoE Architectures
>
> - Existing MoE architectures are primarily applied to the FFN layers of foundation models, focusing on single-modality training or fine-tuning (e.g., text). Our method is the first to introduce a zero expert into the MoE LoRA framework to address catastrophic forgetting in multimodal learning. Therefore, a direct comparison to prior MoE approaches is not straightforward.

---

> ### Author Response · Authors · 2025-08-05
> **Dear Reviewer C5Mg**
>
> As the discussion period draws to a close, we would appreciate it if you could let us know whether our rebuttal has addressed your concerns and if any additional clarifications would help. If you feel your points have been satisfactorily resolved, we would appreciate it if you could consider increasing your score.
>
> Sincerely,
> Authors

---

> > ### Author Response · Authors · 2025-08-07
> > **Matters for discussion**
> >
> > Dear Reviewer C5Mg,
> >
> > We hope this message finds you well.
> >
> > We wanted to kindly follow up on our earlier message. As the discussion deadline is approaching, we would greatly appreciate it if you could share your thoughts on whether our rebuttal has addressed your concerns or if further clarification is needed.
> >
> > Best regards,
> >
> > The Authors

---

### Note · Authors · 2025-08-13

Dear Reviewers and ACs,

Thank you for your time and effort throughout the rebuttal and discussion process.

We propose HMVLM, a human motion–vision–language model based on the MoE LoRA framework. Following the rebuttal, we are pleased that **all reviewers (C5Mg, rpyN, sc8H, t2iJ)** valued the **novelty** of our approach, including the zero-expert design for mitigating catastrophic forgetting, unified framework for multiple tasks, and the fine-grained pose representation for improved spatial encoding. Reviewers (C5Mg, sc8H, t2iJ) also praised the **compelling experimental results**, which strongly demonstrate the effectiveness of our method.

In our rebuttal, we provided additional clarifications and analyses to further strengthen the paper:

-  Analysis of the learned gating weights (rpyN, sc8H): We introduced additional quantitative analysis of the gating weights to examine their distribution across different tasks and provided an example to illustrate the importance of the zero expert.

- Ablation on MoE Configuration (C5Mg, rpyN, sc8H): We added an evaluation to clarify the choice of the number of experts and elaborated on the trade-off between single-task and multi-task fine-tuning.

- Body part partitioning (C5Mg, rpyN, sc8H, t2iJ): We clarified the rationale behind body part partitioning (aligned with the T2M dataset annotations) and analyzed the pros and cons of different granularity levels.

These evaluations and clarifications have addressed the reviewers’ concerns, as reflected in the comments during the discussion. All updates will be included in the revised version and are intended to fully address the reviewers' feedback.

Best regards,

The Authors

---

### Decision · Program_Chairs · 2025-09-17

**Decision:**

Accept (poster)

**Comment:**

Despite some presentation density and trade-offs in multi-task performance, the paper offers a useful and novel mechanism, which is zero expert in MoE-LoRA, that reduces catastrophic forgetting in a challenging multimodal setting, plus a thoughtful pose tokenization that improves spatial fidelity. The added analyses during discussion resolved core concerns about how the gating behaves and why the zero expert is preferable to simpler routing.

For the next revision, please include a small table or plot showing the gating network's expert weight distributions for representative prompts, along with a brief description of the gating architecture. It would also help to add a compact cost table for number of experts vs trainable parameters, training time, inference latency. Please deepen the multi-task discussion and explain why part-based tokenization can increase FID yet improve R-precision/MSE. Finally, include matched-backbone baselines and add the cited works on motion pretraining and continual learning.